

# Site-selective generation of lanthanoid binding sites on proteins using 4-fluoro-2,6-dicyanopyridine

Sreelakshmi Mekkattu Tharayil,[1] Mithun C. Mahawaththa,[2] Akiva Feintuch,[3] Ansis Maleckis,[4] Sven Ullrich,[1] Richard Morewood,[1] Thomas Huber,[1] Christoph Nitsche,[1] Daniella Goldfarb,[3] Gottfried Otting[2]

[1] Research School of Chemistry, Australian National University, Canberra, ACT 2601, Australia

[2] ARC Centre of Excellence for Innovations in Peptide & Protein Science, Research School of Chemistry, Australian National University, Canberra, ACT 2601, Australia

[3] Department of Chemical Physics, Weizmann Institute of Science, Rehovot 76100, Israel

[4] Latvian Institute of Organic Synthesis, Aizkraukles 21, LV-1006, Riga, Latvia

*Correspondence to*: Gottfried Otting (gottfried.otting@anu.edu.au)

**Abstract.** The paramagnetism of a lanthanoid tag site-specifically installed on a protein provides a rich source of structural information accessible by nuclear magnetic resonance (NMR) and electron paramagnetic resonance (EPR) spectroscopy. Here we report a lanthanoid tag that reacts selectively with cysteine or selenocysteine with formation of a (seleno)thioether bond
and a short tether between lanthanoid ion and protein backbone. The tag is assembled on the protein in three steps, comprising (i) reaction with 4-fluoro-2,6-dicyanopyridine (FDCP), (ii) reaction of the cyano groups with α-cysteine, penicillamine or β-cysteine to complete the lanthanide chelating moiety and (iii) titration with a lanthanoid ion. FDCP reacts much faster with selenocysteine than cysteine, opening a route for selective tagging in the presence of solvent-exposed cysteine residues. Loaded with $Tb^{3+}$ and $Tm^{3+}$ ions, pseudocontact shifts were observed in protein NMR spectra, confirming that the tag delivers good
immobilisation of the lanthanoid ion relative to the protein, which was also manifested in residual dipolar couplings. Completion of the tag with different 1,2-amino thiol compounds resulted in different magnetic susceptibility tensors. In addition, the tag proved suitable for measuring distance distributions in double electron–electron resonance experiments after titration with $Gd^{3+}$ ions.

## 1 Introduction

Site-specific labelling of a protein with a paramagnetic lanthanoid ion opens many possibilities to investigate the structure of the protein by NMR and EPR spectroscopy. The presence of a paramagnetic lanthanoid ion produces long-range paramagnetic effects that can be observed by NMR spectroscopy. Among these, pseudocontact shifts (PCS) are particularly straightforward to measure and useful for the structural characterisation of protein (Otting, 2010). Using EPR spectroscopy, tags with $Gd^{3+}$ ions have proven highly attractive for measuring nanometre scale distances by double-electron–electron resonance (DEER)
experiments (Giannoulis et al., 2021). The outstanding utility of these experiments has triggered the development of numerous



reagents and strategies for site-specific attachment of lanthanoid ions to proteins and other biological macromolecules (Miao et al., 2022).

To create useful structural information, the ideal lanthanoid binding site should fulfil several criteria. (i) The lanthanoid ion must be held in a defined position relative to the protein to minimize averaging of PCSs and provide accurate

distance information in DEER experiments. This is best achieved by tags that are tied to the protein via a short and rigid linker that nonetheless must not affect the structure of the protein. (ii) The binding site must be site-specific. Most lanthanoid tags are designed for covalent bond formation of a synthetic lanthanoid complex with cysteine thiol groups, but specific attachment to a genetically encoded non-canonical amino acid would be even more attractive. (iii) The tag should be straightforward to synthesize to save costs and effort.

In principle, these criteria could be fulfilled by a non-canonical amino acid designed for directly binding a lanthanoid ion and amenable to incorporation into the polypeptide chain in response to a stop codon. The amino acid 2-amino-3-(8-hydroxyquinolin-3-yl) propanoic acid has been genetically encoded for this purpose, but its presence in proteins leads to quantitative precipitation upon titration with lanthanide ions (Jones et al., 2010). Phosphoserine is another genetically encoded amino acid capable of metal binding and has successfully been used, in conjunction with another negatively charged amino

acid, to generate lanthanoid binding sites in proteins (Mekkattu Tharayil et al., 2021). Unfortunately, the high concentration of negative charges compromises protein stability and expression yields, limiting the approach to highly stable proteins only.

If a longer tether between the protein backbone and the gadolinium ion can be accepted, the non-canonical amino acid *p*-azidophenylalanine can be installed in the target protein site-selectively, followed by covalent attachment of a gadolinium complex in a $Cu^+$-catalysed click reaction (Abdelkader et al., 2015). Unfortunately, many proteins precipitate upon

exposure to the $Cu^+$ catalyst and the synthesis of suitable gadolinium complexes is demanding.

The present work sought to identify a lanthanoid tag that fulfils the criteria of rigidity, affordability and selectivity for the non-canonical amino acid selenocysteine, which can be incorporated into proteins site-selectively by using a photocaged precursor (Welegedara et al., 2018). The tagging approach uses a strategy of chemical assembly on the target protein, which is based on recently introduced conjugation chemistry, where a cyano-pyridine is ligated with a 1,2-aminothiol under

biocompatible conditions (Nitsche et al., 2019). The tag assembly starts by reacting FDCP with cysteine or selenocysteine in a nucleophilic substitution reaction. In the next step, the dicyanopyridine (DCP) moiety is reacted with two molecules of cysteine, penicillamine or β-cysteine to complete the lanthanide binding motif on the protein (Fig. 1).

The capacity to assemble different tags from readily accessible building blocks is of particular interest for protein structure elucidation. Recent results showed that the installation of four or more tags at the same protein site enable high-

resolution structure determinations at selected sites from PCSs only, provided that the tags generate magnetic susceptibility anisotropy tensors of different orientation (Orton et al., 2022).

In the following, we report on the selectivity of the approach, reaction yields and performance in PCS and DEER measurements.



## 2 Experimental procedures

### 2.1 Expression vector construction

Expression vectors were as published previously (Liepinsh et al., 2001; Guignard et al., 2002; Potapov et al., 2010; Yagi et al., 2011; Welegedara et al., 2017 and 2021; Johansen-Leete et al., 2022) or prepared specifically for the present work by cloning the gene between the *Nde*I and *Eco*RI sites of the T7 vector pETMCSI (Neylon et al., 2000). The final plasmids were constructed using RQ-SLIC and mutagenesis experiments were conducted by QuikChange, where both protocols relied on a mutant T4 DNA polymerase (Qi et al., 2019).

### 2.1 Protein expression

Samples were produced of nine different proteins. The proteins were the *E. coli* peptidyl-prolyl *cis-trans* isomerase PpiB, the N-terminal domain of *P. falciparum* Hsp90, rat ERp29 and its cysteine mutants S114C/C157S and G147C/C157S, the SARS-2 main protease, the T237C/T345C mutant of the maltodextrin binding protein (MBP), the Q32C cysteine mutant of B1 immunoglobulin-binding domain of streptococcal protein G (GB1), and the intracellular domain of the p75 neurotrophin receptor (p75[NTR]). All proteins were expressed in *E. coli* BL21 DE3 cells transformed with the requisite plasmid. Standard expressions used 1 L of cell-culture grown in LB medium with 50 μM spectinomycin and 50 μM kanamycin at 37 °C until the $OD_{600}$ value reached 0.6–0.8. Expression was induced with 1 mM IPTG. Afterwards, the culture was grown at room temperature overnight. [15]N labelling was achieved using a modified protocol developed for a fermenter (Klopp et al., 2018). Initially cells were inoculated in 25 mL [15]N minimal medium (6.8 g/L $KH_2PO_4$, 7.1 g/L $Na_2HPO_4$, 0.71 g/L $Na_2SO_4$, 2.0 mL/L 1 M $MgCl_2$, 18 g/L glucose, 2.6 g/L [15]$NH_4Cl$, 0.2 mL/L trace metal mixture) and grown overnight at 37 °C shaking at 220 rpm. The overnight culture was inoculated into 0.5 L [15]N minimal fermenter medium in a Labfors 5 fermenter (Infors, Bottmingen, Switzerland) and grown until $OD_{600}$ reached 12–13, then 9 g glucose and 1.3 g of [15]$NH_4Cl$ were added and expression induced with 1 mM IPTG. After induction, the cultures were grown at 18 °C overnight for protein expression.

The cells were harvested by centrifugation at 5,000 *g* for 15 minutes and lysed by passing twice through a Emulsiflex-C5 Homogenizer (Avestin, Canada). The lysate was centrifuged at 13,000 *g* for 1 h and the filtered supernatant loaded onto a 5 mL Ni-NTA column (GE Healthcare, USA) equilibrated with binding buffer (50 mM Tris-HCl, pH 7.5, 300 mM NaCl, 5 % glycerol). The protein was eluted with elution buffer (binding buffer containing, in addition, 300 mM imidazole) and the fractions were analysed by 12% SDS-PAGE. Subsequently, for the proteins used for NMR measurements, the His6-tag was removed by digestion overnight at 4 °C, using TEV protease added in 100-fold excess in buffer containing 50 mM Tris-HCl, pH 8.0, 300 mM NaCl and 1 mM β-mercaptoethanol.

Calmodulin K148U and MBP T237U/T345U (where U stands for selenocysteine) were produced by cell-free protein synthesis following a published protocol (Welegedara et al., 2021). In the cell-free protein synthesis reaction, cysteine was replaced by selenocysteine and 10 mM DTT were added to keep selenocysteine in the reduced state. Protein purifications used



a 1 mL His GraviTrap TALON column (GE Healthcare, USA) as described above, except that the buffers were supplemented with 1 mM DTT.

Final protein concentrations were determined by measuring the absorbance at 280 nm, using calculated extinction coefficients for the untagged proteins (Pace et al., 1995), which were 11460 $M^{-1}$ $cm^{-1}$ for GB1 Q32C, 27390 $M^{-1}$ $cm^{-1}$ for ERp29 S114C/C157S and ERp29 G147C/C157S, and 67840 $M^{-1}$ $cm^{-1}$ for MBP T237C/T345C and MBP T237U/T345U. The

reaction of DCP with two 1,2-aminothiols leads to a conjugated double-bond system (Fig. 1) with a molar extinction coefficient of 6850 $M^{-1}$ $cm^{-1}$ at 280 nm for DCP-(L-Cys)$_2$ and 5400 $M^{-1}$ $cm^{-1}$ for DCP-(L-pen)$_2$, which needs to be considered when determining the concentrations of tagged proteins by UV absorption.

## 2.2 Synthesis of 4-fluoro-2,6-dicyanopyridine (FDCP) and β-cysteine

FDCP was synthesized from commercially available 4-chloro-2,6-dicyanopyridine (Ambeed, USA) in a single step by heating

with CsF as reported elsewhere (Ullrich et al., 2022). The Supporting Information provides the detailed protocol for the synthesis of (S)-3-amino-4-mercaptobutanoic acid (β-cysteine).

## 2.3 FDCP tagging reaction

50 µM solutions of protein were first incubated with 10 mM DTT for 0.5 h to reduce the cysteines, followed by buffer exchange to reaction buffer (50 mM Tris-HCl, pH 7.5, 300 mM NaCl) to remove DTT. The tagging reaction was performed with 250

µM FDCP and all ligation reactions were performed at 25 °C, incubating overnight to tag cysteine residues and for 10 minutes to tag selenocysteine residues. Afterwards, excess FDCP was removed by buffer exchange with reaction buffer. In the next step, the DCP-tagged protein was reacted with excess 1,2-aminothiol to obtain the final protein with lanthanide-binding tag. Tags were assembled using five different 1,2-aminothiol compounds, including L-cysteine, D-cysteine, L-penicillamine, D-penicillamine and β-cysteine. The reaction conditions involved 0.5 M 1,2-aminothiol compound, 50 µM DCP-tagged protein,

10 mM TCEP, 50 mM Tris-HCl pH 7.5, 300 mM NaCl and 0.5 h incubation at 25 °C (Fig. 1).

## 2.4 Mass spectrometry

Whole protein mass spectrometry was performed using an Elite Hybrid Ion Trap-Orbitrap mass spectrometer (Thermo Scientific, USA) coupled with an UltiMate S4 3000 UHPLC (Thermo Scientific, USA). 7.5 pmol of sample were injected to the mass analyser via an Agilent ZORBAX SB-C3 Rapid Resolution HT Threaded Column (Agilent, USA).

## 2.5 NMR spectrometry

All NMR spectra were recorded at 25 °C, using an 800 MHz Bruker Avance NMR spectrometer. Samples were prepared in 20 mM HEPES buffer, pH 7.0, in 3 mm NMR tubes. 10 % D$_2$O was added to provide a lock signal. 0.1–0.5 mM protein samples were used for 2D [$^{15}$N,$^{1}$H]-HSQC experiments. 10 mM LnCl$_3$ stock solutions were used to titrate NMR samples.




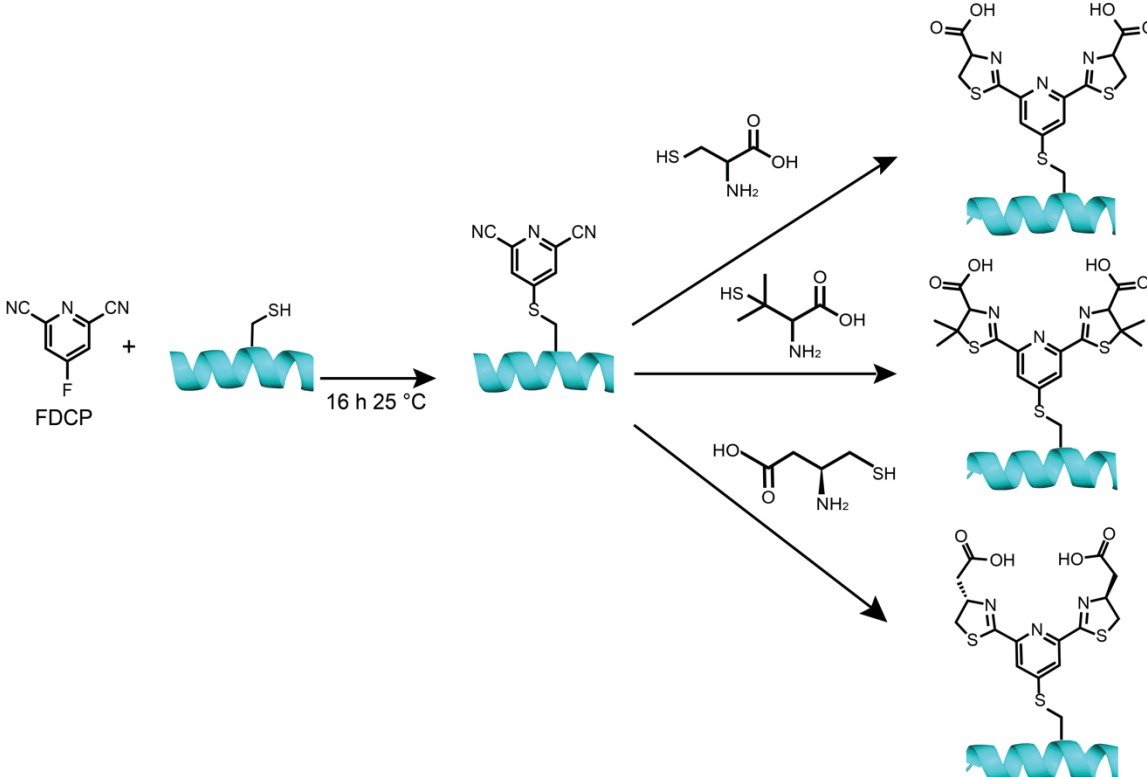

**Figure 1. Construction of lanthanide binding motifs on a protein using FDCP.** Initially, a solvent-exposed cysteine or selenocysteine residue on the protein is reacted with FDCP. Typical reaction conditions involve incubating the protein at a concentration of 50 µM with 250 µM FDCP at 25 ℃ and pH 7.5 for 16 h (10 minutes for selenocysteine) to install the DCP
tag on the protein. Prior to the tagging reaction, the reduced state of the cysteine thiol group was ensured by treating the protein with 10 mM DTT for 0.5 h followed by extensive buffer exchange to remove DTT. A second step assembles the final lanthanide-binding motif by reacting the FDCP-tagged protein for 0.5 h at 25 ℃ and pH 7.5 in the presence of 10 mM TCEP with either (top) 0.5 M L- or D-cysteine, (middle) 0.5 M L- or D-penicillamine or (bottom) 0.5 M (S)-3-amino-4-mercaptobutanoic acid (β-cysteine).

### 135  2.6 PCS measurements and Δχ-tensor fitting

Pseudocontact shifts (PCS) were measured in ppm as the difference in amide proton chemical shift between the paramagnetic

and diamagnetic NMR spectrum. PCSs were used to determine the position and orientation of the Δχ tensor of the paramagnetic

ions relative to the protein structure. Fitting of Δχ tensors was performed using the program Paramagpy (Orton et al., 2020).

### 2.7 Residual dipolar coupling measurements

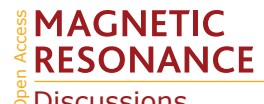

Residual dipolar couplings (RDCs) of one-bond $^{1}$H-$^{15}$N couplings were measured for GB1 Q32C DCP-(L-Cys)$_2$ and GB1 Q32C DCP-(D-pen)$_2$ loaded with Tb$^{3+}$ ions, using the IPAP [$^{15}$N,$^{1}$H]-HSQC experiment (Ottiger et al., 1997) with $t_{1max}$ = 75 ms. The RDCs were calculated as the one-bond splittings measured for GB1 with the paramagnetic Tb$^{3+}$ tag minus the corresponding values measured with the diamagnetic Y$^{3+}$ tag. The RDCs were used as input for the program Paramagpy (Orton et al., 2020) to fit alignment tensors and translate them into Δχ tensor parameters (leaving the metal position undetermined).

## 2.8 DEER measurements

In order to check whether FDCP can be used as a tool for measuring DEER distances, the protein samples were buffer exchanged to 50 mM Tris-HCl in D$_2$O pD 7.5 (uncorrected pH metre reading) and concentrated to 100 μM following the treatment with excess of L-cysteine or β-cysteine. Gadolinium was added from a 2.5 mM stock solution of GdCl$_3$. Perdeuterated glycerol was added to a final concentration of 20% (v/v) to reach a final protein concentration of 0.1 mM.

All pulsed EPR measurements were carried out at 10 K on a home-built W-band (95 GHz) spectrometer (Goldfarb et al., 2008) equipped with an arbitrary waveform generator (Bahrenberg et al., 2017). Echo-detected EPR (ED-EPR) spectra were recorded using the $\pi/2 - \tau - \pi - \tau -$ echo sequence, with a two-step phase cycle (0,$\pi$) on the first $\pi/2$ pulse, while keeping $\tau$ = 500 ns and sweeping the magnetic field. The durations of the $\pi/2$ and $\pi$ pulses were 15/30 ns, respectively.

DEER measurements employed the standard four-pulse DEER sequence $\pi/2_{vobs} - \tau_1 - \pi_{vobs} - (\tau_1+t) - \pi_{vpump} - (\tau_2-t) - \pi_{vobs} - \tau_2 -$ echo (Pannier et al., 2000). A chirp pump pulse monitoring the echo intensity with increasing delay $t$ and an eight-step phase cycle were applied. The pump pulses with a duration of 128 ns were set to the central transition to cover the range 94.9–95.05 GHz (150 MHz bandwidth) and the observed pulses were set to 94.85 GHz with pulse durations $\pi/2$ ($\pi$) = 15 (30) ns. The maximum of the Gd$^{3+}$ spectrum was set to 94.9 GHz. The repetition
delay was 200 μs and the evolution time depended on the sample. The time domain DEER data were analyzed using the DeerAnalysis2022 software package (Jeschke et al., 2006). The analysis was carried out with the Tikhonov regularization. The background decay was fitted with a dimension of three. Default values were used for the validation process including noise addition. DEER traces were also analyzed with DeerNet (Worswick et al., 2018) through the DeerAnalysis interface.

## 2.9 Modelling

The experimental DEER distance distributions were compared with distance distributions obtained by crafting models of the Gd$^{3+}$ tags onto crystal structures of the human ERp29 dimer (PDB ID: 2QC7; (Barak et al., 2009)) and MBP (PDB ID: 1OMP; (Sharff et al., 1992)) generating rotamer libraries of the tags with the program PyParaTools as described previously (Stanton-Cook et al., 2014; Welegedara et al., 2021). To predict the Gd$^{3+}$–Gd$^{3+}$ distances, the tags were crafted onto cysteine residues
placed at positions 114 or 147 (for ERp29) and 237 and 345 (for MBP). The rotamer libraries were generated for each tag allowing the χ$_1$ angle to vary by ±30° around the staggered rotamers, while the χ$_2$ and χ$_3$ angles, which precede and follow the



sulfur atom respectively (Fig. S11), were allowed to rotate freely. PyParaTools predicts distance distributions by assuming equal population of each tag conformation that is free of van der Waals clashes between tag and protein.

## 3 Results

### 3.1 Reactivity of FDCP towards cysteine and selenocysteine residues

Cysteine thiol groups react with FDCP in a nucleophilic substitution reaction on the pyridine ring (Fig. 1). To maintain the thiol groups in the reduced state, samples were incubated with DTT and washed with reaction buffer (50 mM Tris-HCl, pH 7.5, 300 mM NaCl) prior to incubation with FDCP. Initially, the reaction conditions were optimized for the mutant GB1 Q32C, where mass spectrometry indicated complete reaction following incubation overnight at room temperature (Fig. S1). To test whether selenocysteine is more reactive towards FDCP, calmodulin K148U, where U stands for selenocysteine, was subjected to the same reaction conditions, except that the reaction was conducted in the presence of 1 mM DTT to maintain selenocysteine in the reduced state. The reaction was found to complete within 10 minutes, highlighting the selectivity of FDCP for selenocysteine. There was no evidence of DTT competing with FDCP for ligation with selenocysteine (Fig. S2).

The role of solvent exposure in the tagging reaction with FDCP was explored using five different proteins containing cysteine residues of varying solvent accessibility. The protein PpiB contains two buried cysteine residues (Cys31 and Cys121). Mass spectrometric analysis indicated that these cysteine residues remain unaffected by the tagging reaction with FDCP (Fig. S3a). The N-terminal domain of *P. falciparum* Hsp90 contains a cysteine residue in position 209, which is near the surface but, according to the crystal structure 3K60 (Corbett et al., 2010), protected from solvent access by a glutamate side chain. The overnight tagging reaction resulted in no tagging (Fig. S3b). The homodimeric rat protein ERp29 contains a single cysteine residue in position 157, which is substantially but not fully solvent exposed. It resulted in about 15 % tagging yield (Fig. S3c). In contrast, ERp29 mutated to contain a cysteine residue in position 114 while the natural cysteine was mutated to serine (mutant S114C/C157S) showed high yield of tagging of Cys114 (Fig. S3d). Residue 114 is very highly solvent-exposed in the NMR structure (1G7E; Liepinsh et al., 2001) and the crystal structure 2QC7 of the human homologue (Barak et al., 2009). The SARS-2 main protease is a cysteine protease containing twelve cysteine residues. Again, FDCP resulted in very little tagging. A small mass spectrometric peak suggested partial tagging of a single cysteine residue (Fig. S3e). In principle, the active-site cysteine is partially solvent-exposed but the crystal structure (6Y2E; Zhang et al., 2020) indicates that three other cysteine residues are at least as accessible. In contrast, the two highly solvent-exposed cysteine residues of the intracellular domain of p75[NTR] (Cys379 and Cys416) both were readily tagged (Fig. S3f). These results indicate that complete ligation of FDCP with cysteines can readily be achieved in an overnight reaction, provided that the cysteine thiol groups are highly solvent-exposed.

### 3.2 Reaction conditions for tag assembly

Following tagging with FDCP, the DCP-tagged proteins were reacted with either cysteine or penicillamine to complete the lanthanoid chelating moiety of the tag (Fig. 1). The necessary reaction conditions were optimised using GB1 Q32C tagged



with DCP. 0.5 M cysteine was found to complete the reaction with both cyano groups in half an hour (Fig. S1). To prevent the precipitation of cystine, the reaction buffer was supplemented with 10 mM TCEP, and the pH adjusted to 7.5 if required.

High-salt conditions did not accelerate the initial reaction of protein with FDCP, but the following reaction of the cyano groups with cysteine was significantly aided by the presence of salt. For example, none of the cyano groups of the DCP tag reacted in an overnight reaction with 10 mM cysteine and no salt, whereas one of the cyano groups reacted when incubating overnight with 10 mM cysteine in the presence of 300 mM NaCl. Completion of the reaction of both cyano groups in less than one hour required high concentrations of both salt and cysteine (or penicillamine).

## 210    3.3 Pseudocontact shifts and RDCs with DCP tags

The potential of the DCP-(L-Cys)$_2$ tag as a lanthanoid tag for the generation of PCSs was investigated using the $^{15}$N-labelled GB1 mutant Q32C. Titration of the protein tagged with DCP-(L-Cys)$_2$ with TbCl$_3$ to a metal-to-protein ratio of about 0.6:1 generated PCSs of up to 1.5 ppm in [$^{15}$N,$^1$H]-HSQC spectra (Table S1, Fig. 2a). At this titration ratio, cross-peaks of the metal-bound protein co-existed with weak cross-peaks of the metal-free protein, indicating slow exchange between proteins with and

without metal ion. The experimental PCSs allowed fitting of Δχ tensors with good (i.e., low) quality factors, indicating structural conservation of the protein and little mobility of the tag (Table 1).

        Metal-induced aggregation led to protein precipitation at metal-to-protein ratios much greater than 0.6:1. To maintain narrow NMR signals and avoid precipitation, subsequent work focused on samples prepared with metal-to-protein ratios of about 0.6:1.

To explore the potential to vary the chemical and magnetic properties of the tag by changing the 1,2-aminothiol reagent used, we also tested the use of D-cysteine as well as L- and D-penicillamine to complete the DCP tag (Fig. 1). Indeed, the Δχ tensors obtained with DCP-(D-Cys)$_2$ differed in sign, magnitude and orientation from those obtained with DCP-(L-Cys)$_2$ (Table 1 and Fig. 2b), and tag assembly with L- and D-penicillamine again resulted in different Δχ tensors (Table 1 and Fig. 3). As the chemical structure of penicillamine differs from cysteine by two methyl groups in place of the β-hydrogens,

penicillamine may be expected to generate a more rigid lanthanide-complexation geometry than cysteine. Some of the Δχ tensors obtained with the tags constructed with penicillamine were quite large, but as the distances between the fitted metal position and the site of tag attachment (Cys32) were also increased, this observation may be attributed to tag flexibility rather than intrinsically large Δχ tensors. To test this hypothesis, we also measured $^1D_{HN}$ RDCs for two of the samples, GB1 Q32C DCP-(L-Cys)$_2$-Tb$^{3+}$ and GB1 Q32C DCP-(D-pen)$_2$-Tb$^{3+}$. As expected for a poorly determined metal position, the alignment

tensors were significantly smaller than predicted from the Δχ tensor determined from PCSs (Table 1). Nonetheless, the Δχ tensor fits of the PCSs of all tags delivered very good $Q$ factors despite their mobile character. Furthermore, the z-axes of the tensors obtained with the different tags diverted significantly, with the closest alignment (observed between the DCP-(D-Cys)$_2$ and DCP-(L-pen)$_2$ tags) featuring an angle of 11° between the respective z-axes. The family of tags of the present work

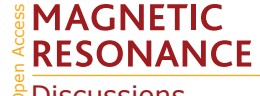

thus presents a good basis for generating largely independent Δχ tensors as required for obtaining structural information from multiple tags attached to a single site (Orton et al., 2022).

**Figure 2. Pseudocontact shifts and Δχ tensors in the protein GB1 Q32C with DCP-Cys tag.** (a) Superimposition of [$^{15}$N,$^{1}$H]-HSQC spectra of an 0.3 mM solution of GB1 Q32C tagged with DCP-(L-Cys)$_2$ and loaded with either Tb$^{3+}$ (red cross peaks), Tm$^{3+}$ (blue) or Y$^{3+}$ (black). The metal ions were provided in a metal-to-tag ratio of about 0.6:1. Lines connect





corresponding cross-peaks observed with diamagnetic and paramagnetic metal ions. (b) Same as (a), but for GB1 Q32C tagged with DCP-(D-Cys)₂. (c) Correlation between back-calculated and experimental PCSs of GB1 Q32C tagged with DCP-(L-Cys)₂. Red and blue points correspond to the PCSs of $Tb^{3+}$ and $Tm^{3+}$, respectively. (d) Same as (c), but for the tag assembled with D-Cys. (e) PCS isosurfaces of ±1 ppm determined by the Δχ tensor of $Tb^{3+}$ bound to the DCP-(L-Cys)₂ tag and plotted on a ribbon representation of GB1 Q32C. Positive and negative isosurfaces are shown in blue and red, respectively. The side chain of

Cys32 is highlighted with sticks and the metal position obtained by the Δχ tensor fit is identified by a ball. (f) Same as (e), but for $Tm^{3+}$. (g) Isosurfaces for the tag constructed with D-Cys and loaded with $Tb^{3+}$. (h) Same as (g), but for $Tm^{3+}$.



**Figure 3. Pseudocontact shifts and Δχ tensors in the protein GB1 Q32C with DCP-penicillamine (DCP-pen₂) tag.** (a) Superimposition of [$^{15}$N,$^1$H]-HSQC spectra. Spectra recorded of an 0.3 mM solution of GB1 Q32C tagged with DCP-(L-pen)₂ and loaded with either Tb$^{3+}$ (red), Tm$^{3+}$ (blue) or Y$^{3+}$ (black). (b) Same as (a), but for GB1 Q32C tagged with DCP-(D-pen)₂. (c) Correlation between back-calculated and experimental PCSs (Tb$^{3+}$: red; Tm$^{3+}$: blue) of GB1 Q32C tagged with DCP-(L-pen)₂. (d) Same as (c), but for the tag assembled with D-pen. (e) PCS isosurfaces of +1 ppm (blue) and -1 ppm (red) in GB1 Q32C with DCP-(L-pen)₂ tag and Tb$^{3+}$ ion. The side chain of Cys32 is highlighted by a stick representation and a ball marks the metal position obtained from the Δχ-tensor fit. (f) Same as (e), but for Tm$^{3+}$. (g) Isosurfaces for the tag constructed with D-pen and loaded with Tb$^{3+}$. (h) Same as (g), but for Tm$^{3+}$.

**Table 1.** Δχ-tensor parameters of GB1 Q32C with DCP-Cys₂ or DCP-pen₂ tags and titrated with Tb$^{3+}$ and Tm$^{3+}$ ions.[1]

| Protein construct | $\Delta\chi_{ax}$ ($10^{-32}$ m³) | $\Delta\chi_{rh}$ ($10^{-32}$ m³) | $x$ (Å) | $y$ (Å) | $z$ (Å) | α (°) | β (°) | γ (°) | $d$ (Å) | $Q$ |
|---|---|---|---|---|---|---|---|---|---|---|
| GB1 Q32C DCP-(L-Cys)₂-Tb | -21.6 | -2.0 | 35.958 | 35.881 | 19.408 | 34 | 39 | 50 | 10.0 | 0.05 |
| GB1 Q32C DCP-(L-Cys)₂-Tm | 16.2 | 4.7 | 35.958 | 35.881 | 19.408 | 14 | 37 | 37 | 10.0 | 0.08 |
| GB1 Q32C DCP-(D-Cys)₂-Tb | 6.8 | 1.2 | 34.322 | 34.833 | 19.466 | 179 | 114 | 171 | 8.1 | 0.06 |
| GB1 Q32C DCP-(D-Cys)₂-Tm | -3.4 | -0.5 | 34.322 | 34.833 | 19.466 | 179 | 112 | 141 | 8.1 | 0.09 |
| GB1 Q32C DCP-(L-pen)₂-Tb | -37.8 | -13.6 | 32.667 | 39.609 | 19.491 | 14 | 54 | 68 | 10.3 | 0.04 |
| GB1 Q32C DCP-(L-pen)₂-Tm | 17.7 | 3.9 | 32.667 | 39.609 | 19.491 | 10 | 64 | 65 | 10.3 | 0.08 |
| GB1 Q32C DCP-(D-pen)₂-Tb | 71.5 | 37.7 | 26.888 | 36.157 | 7.699 | 168 | 69 | 106 | 11.9 | 0.03 |
| GB1 Q32C DCP-(D-pen)₂-Tm | -36.4 | -20.6 | 26.888 | 36.157 | 7.699 | 167 | 70 | 107 | 11.9 | 0.06 |
| GB1 Q32C DCP-(D-pen)₂-Tb[2] | 15.2 | 3.9 | | | | 168 | 77 | 28 | | 0.26 |
| GB1 Q32C DCP-(L-Cys)₂-Tb[2] | 8.1 | 5.3 | | | | 0 | 51 | 111 | | 0.44 |

[1] The coordinates of the paramagnetic centre and the Euler angles describing the orientation of the tensor refer to the protein coordinates 1PGA (Gallagher et al., 1994). $d$ is the distance between the β-carbon of residue 32 and the fitted metal position. The fits assumed a common metal position for tags loaded with Tb$^{3+}$ and Tm$^{3+}$ ions. Quality factors $Q$ were calculated as the ratio of the root-mean-square deviation between experimental (Tables S1 and S2) and back-calculated PCSs and the root-mean-square of the experimental PCSs.

[2] Tensor parameters obtained from $^1D_{HN}$ RDC measurements (Table S3). The alignment tensor $A$ was expressed in Δχ tensor units using $\Delta\chi_{ax,rh} = (15\mu_0 k_B T/B_0^2)A_{ax,rh}$, where $\mu_0$ is the induction constant, $k_B$ the Boltzmann constant, $T$ the absolute temperature and $B_0$ the magnetic field strength (Bertini et al., 2002).

### 3.4 DCP tags for distance measurements by DEER experiments

The small number of rotatable bonds in the DCP-cysteine and DCP-penicillamine conjugates suggest that the position of the metal ion may be better defined than in tags with longer and less rigid tethers. To test this hypothesis independent of NMR



experiments, we prepared samples with two tags, loaded them with $Gd^{3+}$ ions and conducted DEER experiments, where immobile tags can deliver narrow distributions of $Gd^{3+}$–$Gd^{3+}$ distances, whereas flexible tags invariably result in broad distance distributions (Welegedara et al., 2017 and 2021; Prokopiou et al., 2018; Widder et al., 2019). The experiments were conducted on a W-band EPR spectrometer following titration of the samples with $GdCl_3$. To start with, we assembled the DCP-(L-Cys)$_2$

tag on the double-mutant G147C/C157S of rat ERp29, which is a homodimer. In addition, we produced the cysteine and selenocysteine mutants of the maltose binding protein, MBP T237C/T345C and MBP T237U/T345U, respectively. Mass spectrometric analysis confirmed complete reaction of all the proteins with FDCP and the subsequent completion of the tags by reaction with 0.5 M cysteine also proceeded in near-quantitative yield (Fig. S4). Samples targeting equimolar ratios of $Gd^{3+}$ ion to tag contained about 20 % excess of $Gd^{3+}$ ions, as the UV absorption of the tag had accidentally been neglected, leading

to an overestimation of the protein concentration.

Figure 4 shows the DEER results obtained for ERp29 G147C/C157S, MBP T237U/T345U and MBP T237C/T345C with a 1.2:1 ratio of $Gd^{3+}$ ion to tag and MBP T237C/T345C with a metal-to-tag ratio of 0.6:1. At the 1.2:1 $Gd^{3+}$ ion-to-tag ratio, a narrow distance distribution was observed for ERp29 G147C/C157S, but the distance distribution was broader for MBP T237U/T345U. Unexpectedly, MBP T237C/T345C at the same metal ion-to-tag ratio yielded a distance distribution with two peaks, whereas the same mutant with a sub-stoichiometric ratio of $Gd^{3+}$ ion to tag of 0.6:1 yielded a narrow distance

distribution without the additional peak at short distance. The additional peak suggests the existence of an alternative metal binding site that is populated by the excess of $Gd^{3+}$ ions and depleted at lower $Gd^{3+}$ ion concentrations as indicated by the distance distribution observed in Fig. 4d. The narrow distance distribution of Fig. 4d may be aided by dimerization, if DCP-(L-Cys)$_2$ tags of different protein molecules share a single lanthanoid ion as suggested by our NMR results. If this were the case, the positions of the tags could be greatly restricted relative to the protein. Notably, however, the second peak maximum

was prominent only in the case of the MBP T237C/T345C mutant, but not in the case of MBP T237U/T345U (Fig. 4b), which is difficult to attribute to differences between selenocysteine and cysteine, although the selenium atom is associated with slightly longer bonds and smaller bond angles.

To guard against coordination of the metal ion by two tag molecules, we reacted the DCP tags with β-cysteine instead

of L-cysteine to produce the DCP-(β-Cys)$_2$ tag (Fig. 1c). Figure 5 shows the DEER results obtained with DCP-(β-Cys)$_2$ tags on the mutants ERp29 S114C/C157S and MBP T237C/T345C, following titration of the tags with $Gd^{3+}$ ions. Despite the inaccurate titration ratio, MBP T237C/T345C gave a narrow distance distribution with the maximum at the expected distance (Fig. 5b), with no evidence for a second lanthanoid binding site.

In summary, all samples made with DCP tags featured exceedingly low modulation depths and this was particularly

the case for the tags prepared with β-cysteine. In part, this result may be attributed to the relatively broad central transition observed in the ED-EPR spectrum, which is not favourable for DEER experiments. For example, the W-band echo-detected EPR spectra of MBP T237U/T345U and MBP T237C/T345C tagged with DCP-(L-Cys)$_2$-Gd, and ERp29 S114C/C157S labelled with DCP-(β-Cys)$_2$-Gd gave a relative broad central transition with a full width at half-height of 6–7.6 mT (Fig. S6).



For comparison, the BrPy-DO3A-Gd and DOTA-maleimide-Gd tags feature central transition linewidths of 4.5 and 1.5 mT,
respectively (Giannoulis et al., 2021).

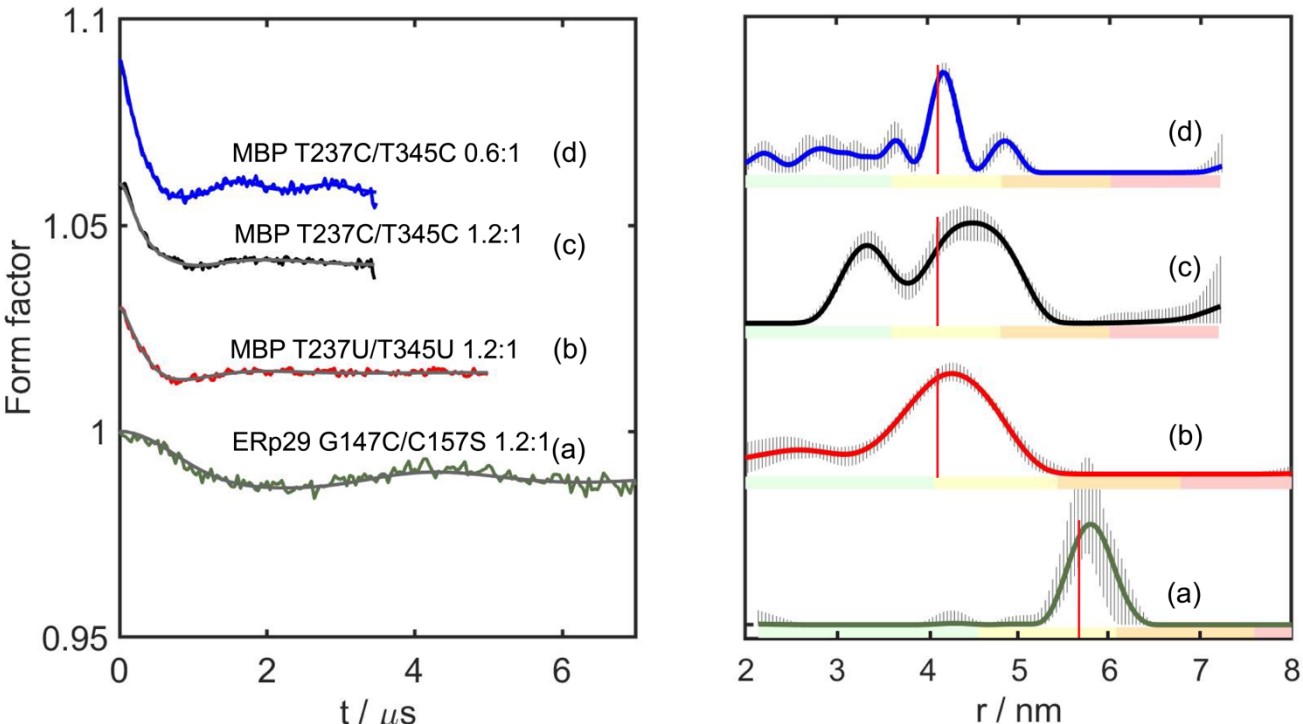

**Figure 4. DEER distance measurements with DCP-(L-Cys)₂-Gd tags.** The left panel shows the form factors after
background correction, where the vertical axis plots the normalized echo intensity and the red line corresponds to the fitted
trace using the distance distribution calculated by DeerAnalysis2022 (Jeschke et al., 2006) and displayed in the centre panel.
The right panel shows the corresponding distance distributions predicted with the program PyParaTools (Fig. S12; Stanton-
Cook et al., 2014). The data are annotated with the names of the proteins and the molar ratios of $Gd^{3+}$ ions to DCP-Cys₂ tagging
sites. (a) ERp29 G147C/C157S with a metal-to-tag ratio of 1.2:1. (b) MBP T237U/T345U with a metal-to-tag ratio of 1.2:1.
(c) MBP T237C/T345C with a metal to tag ratio of 1.2:1. (d) Same as (c), but with a metal to tag ratio of 0.6:1. The red vertical
lines indicate the maxima of the modelled distance distributions. The colour bar underneath the distance distributions indicates
the reliability regions as defined in DeerAnalysis and determined by the DEER evolution time (green: the shape of the distance
distribution is reliable; yellow: the mean distance and distribution width are reliable; orange: the mean distance is reliable; red:
unreliable long-range distances). The solid lines represent the distributions with the smallest root mean square deviation in
relation to the experimental data. The striped regions indicate the range of alternative distributions (±2 times the standard
deviation). The primary DEER data are shown in Fig. S7 and the distance analysis by DeerNet (Worswick et al., 2018) is given
in Fig. S8.






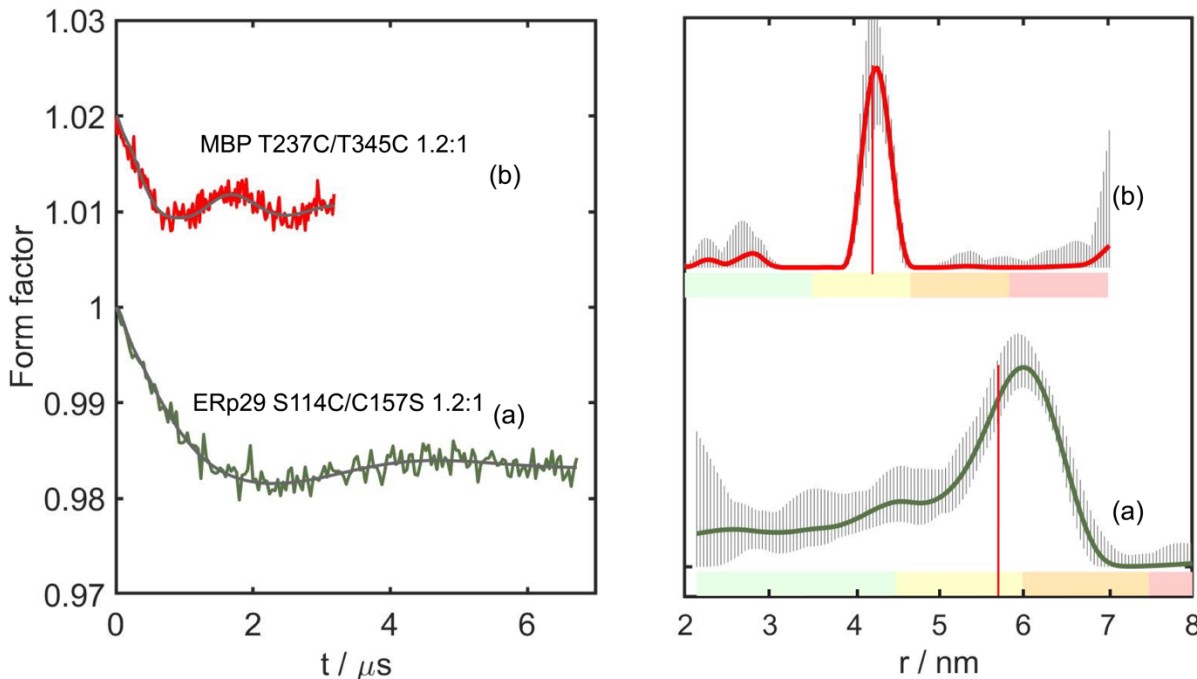

**Figure 5. DEER distance measurements with DCP-(β-Cys)₂-Gd tags.** As in Fig. 4, the left and right panels show the form factors after background subtraction and the distance distribution calculated by DeerAnalysis2018 (Jeschke et al., 2006). All samples were recorded with 1.2:1 Gd$^{3+}$ ion-to-tag ratio. (a) ERp29 S114C/C157S. (b) MBP T237C/T345C. The red vertical lines indicate the maxima of the modelled distance distributions modelled with the program PyParaTools (Fig. S13; Stanton-Cook et al., 2014). The uncertainties in the distance distributions and ranges of alternative distributions are indicated as in Fig. 4. The primary DEER data are shown in Fig. S9 and the distance analysis of (b) by DeerNet (Worswick et al., 2018) in Fig. S10.


**4 Discussion**

Despite numerous designs of lanthanoid tags and tagging strategies published over the past two decades (Nitsche and Otting, 2017; Su and Chen, 2019; Joss and Häussinger, 2019; Miao et al., 2022) none of the currently available designs simultaneously satisfies all criteria that would make a perfect lanthanoid tag, including rigid attachment to the target molecule via a short tether without causing structural perturbations, chemical selectivity, ease of chemical synthesis and affordability.

The present work assessed a new approach, where the final tag is chemically assembled on the target molecule from different readily accessible building blocks, thus providing easy access to multiple different variants with different paramagnetic properties. Chemical assembly of lanthanide tags on cysteine residues has been proposed previously using





pnictogens as mediator between the thiol groups of cysteine and tag, but the pnictogen system depends on a carefully

constructed di-cysteine motif, where both cysteine residues need to present their thiol groups in a suitable geometry, as well as assistance by other amino acid side chains of the protein to immobilise the lanthanide ion sufficiently to enable PCS measurements (Nitsche et al., 2017). The approach of the present work is much more straightforward and general, and it enables the construction of a series of different tags that produce different $\Delta\chi$ tensors as required for high-resolution structure analysis of specific sites of interest in a protein (Orton et al., 2022).

FDCP proved to deliver remarkable selectivity for solvent-exposed cysteine residues. For example, FDCP barely reacted with the SARS-2 main protease, which is a cysteine protease where at least four of the twelve cysteine residues are quite solvent-exposed. Similarly, PpiB, which contains two buried cysteine residues, proved completely unreactive towards FDCP, whereas these cysteine residues are known to react readily with maleimide tags under the same conditions (E. H. Abdelkader, personal communication). This observation agrees with the greater spatial demands of a reaction involving a

nucleophilic substitution than the addition to an alkene as in, e.g., maleimide tags. In contrast, highly solvent-exposed cysteine thiol groups readily deliver complete reaction yields overnight at room temperature and neutral pH. FDCP thus enables selective targeting of a single highly solvent-exposed cysteine residue without having to mutate numerous native cysteine residues to unreactive amino acids like serine or valine.

       Importantly, the reaction of FDCP with selenocysteine was complete within 10 minutes, indicating that selective

tagging of selenocysteine can be achieved in the presence of solvent-exposed cysteine residues. A mutant aminoacyl-tRNA synthetase has been shown to install photocaged selenocysteine in proteins in response to an amber stop codon, providing a generally applicable route to genetic encoding of selenocysteine (Welegedara et al., 2018). Only few lanthanoid binding tags are suitable for forming stable ligation products with selenocysteine (Wu et al., 2017; Herath et al., 2021). Work is in progress to increase the protein yields with photocaged selenocysteine, as this would open a most attractive and widely applicable route

to site-specific tagging of proteins with a short tether.

       While the reactivity of FDCP is high with solvent-exposed cysteine or selenocysteine residues owing to the electron withdrawing effect of the cyano groups, the subsequent reaction of the DCP-labelled protein with cysteine or other 1,2-aminothiols is much slower. We achieved acceptable reaction rates and complete yields by using a generous excess of the β-amino thiol, e.g., 0.5 M cysteine. At concentrations below 100 mM, only one of the cyano groups tended to react with cysteine

derivatives. More facile reaction rates have been observed in reactions involving with 1,2-aminothiol compounds without a charged carboxylate group (Morewood et al., 2021; Patil et al., 2021).

       The ligation of DCP with two cysteine, penicillamine or β-cysteine molecules creates a binding site for lanthanoid ions. Modelling of the complexes indicates, however, that the nitrogen atoms and carboxyl oxygens cannot be positioned to simultaneously contact the metal ion without breaking the conjugated double-bond system of the pyridine and thiazoline rings

(Fig. 1 and S10). The suboptimal coordination geometry may explain why multiple crystallisation attempts of the DCP-(L-Cys)$_2$ complex with $Y^{3+}$ were unsuccessful. Although the titration of DCP-(L-Cys)$_2$ with $YCl_3$ was accompanied by significant changes in chemical shifts in the [1]H-NMR spectrum and the exchange between the free ligand and the complex was slow on



the time scale of chemical shifts, we were unable to determine lanthanoid binding affinities by NMR. In a titration experiment of DCP-(L-Cys)$_2$ with YCl$_3$, excess of metal ion beyond a ratio of 0.5:1 of metal to DCP-(L-Cys)$_2$ did not lead to further

changes of the spectrum, suggesting formation of a stable complex, where a single Y$^{3+}$ ion is coordinated by two DCP-(L-Cys)$_2$ molecules. This not only complicates the analysis but also suggests that a protein with DCP-(L-Cys)$_2$ tag can likewise dimerize by virtue of two DCP-(L-Cys)$_2$ tags sharing a single lanthanoid ion. It is tempting to assume that dimerization could contribute to the more uniform and narrower distance distribution obtained for MBP T237C/T345C with DCP-(L-Cys)$_2$-Gd tags for sub-stoichiometric rather than near-equimolar Gd$^{3+}$ ion-to-tag ratios (Fig. 4c and d), but the distance distribution

obtained with the selenocysteine version of the construct for the same Gd$^{3+}$ ion-to-tag ratio (Fig. 4b) argues against this interpretation. An alternative explanation could be the difficulty to accurately titrate small sample volumes with GdCl$_3$.

The attempt to break potential dimerization by reacting DCP with β-cysteine instead of cysteine or penicillamine yielded a tag that resulted in loss of NMR signals as soon as paramagnetic lanthanoid ions were added. Therefore, no PCSs could be observed. Using the DCP-(β-Cys)$_2$-Gd tags in DEER experiments indicated that these tags do bind lanthanoid ions

and the distance distribution obtained for MBP T237C/T345C was remarkably narrow and free of multiple maxima for the expected near-equimolar Gd$^{3+}$ ion-to-tag ratio (Fig. 5).

All DEER measurements featured low modulation depths for all DCP-1,2-aminothiol-Gd tags, which can be attributed to the unfavourably broad width of the central transition in the ED-EPR spectrum (Fig. S6) compared with previously established gadolinium tags (Garbuio et al., 2013; Collauto et al., 2016; Shah et al., 2019; Herath et al., 2021), which makes

DEER measurements challenging. Furthermore, over-titration or under-titration of the tags with Gd$^{3+}$ ions can also reduce the modulation depths and inaccurate titration ratios arose from ignoring the extinction coefficients of the tags when measuring the protein concentrations by their absorption at 280 nm. In view of the low modulation depths detected in all DEER experiments and the sensitivity of the distance distributions to the actual titration ratio of Gd$^{3+}$ ion to protein, we conclude that these tags are less attractive for DEER distance measurements.

Regarding PCSs, the present work established a new set of tags that are inexpensive, small and relatively rigid, suitable for installation on a highly solvent-exposed cysteine thiol group and capable of delivering Δχ tensor fits with good quality factors. Furthermore, constructing the tags with cysteine or penicillamine generated different PCSs and Δχ tensors of different orientations as required for site-specific structure analysis by PCSs (Orton et al., 2022). The titrations with paramagnetic lanthanoid ions tended to result in broader NMR signals of the paramagnetic species before the complete

disappearance of the signals of the free protein. Therefore, the NMR spectra with the narrowest signals were obtained using a substoichiometric lanthanoid-to-tag ratio, which inferred the simultaneous presence of NMR cross-peaks from paramagnetic and diamagnetic species. The greater complexity of the resulting spectra can be addressed by NMR spectra of higher dimensionality in the case of larger proteins (Orton et al., 2018).

Double-arm tags are known to deliver predictable metal positions and Δχ tensor orientations relative to the protein

(Keizers et al., 2008), whereas attachment to a single cysteine residue results in unpredictable tag orientations. For the DCP

tags of the present work, the limited rigidity of attachment arguably presents an advantage, as it allows generating different $\Delta\chi$ tensor orientations by subtle changes such as completing the tag with 1,2-aminothiols of different chiralities or with different substituents. PCSs obtained with mobile tags are known to deliver excellent structural information so long as an effective $\Delta\chi$ tensor can be fitted that explains the PCSs near the site of interest (Shishmarev and Otting, 2013).

**5 Conclusions**

The present study shows that significantly different $\Delta\chi$-tensor orientations can be obtained by transforming the DCP tags with 1,2-aminothiols of different chirality and with different chemical groups. DCP-tag assembly thus establishes a highly attractive concept for producing the multiple different tensor orientations required to determine local protein structure by multiple PCSs (Orton et al., 2022) or analyse protein motions by RDCs (Tolman et al., 2001; Peti et al., 2002; Vögeli et al., 2008). Although

FDCP is currently not yet commercially available, its synthesis is straightforward and inexpensive. Finally, the much greater reactivity of FDCP with selenocysteine compared to cysteine suggests that selective tagging of selenocysteine can readily be achieved in the presence of solvent-exposed cysteine residues, which is of great interest for protein studies both by NMR and EPR spectroscopy. The flexibility associated with selective tag assembly encourages the search for further tags constructed by this principle.


**Data availability.** The NMR and EPR data are available at https://doi.org/10.5281/zenodo.6585976 and https://doi.org/10.5281/zenodo.6579184, respectively.

**Supplement.** The supplement related to this article is available online at:


**Author contributions.** SMT prepared the EPR samples, calculated distance distributions and drafted the first version of the manuscript, MCM initiated the project, produced the protein samples for NMR and performed the NMR experiments, resonance assignments and $\Delta\chi$ tensor fits, AF performed the EPR measurements and analysis, AM synthesized $\beta$-cysteine, SU synthesized FDCP, RM synthesized DCP-(L-Cys)$_2$ for crystallisation, TH contributed the concept of different tag assemblies,

CN supervised the chemical synthesis at the ANU, DG supervised the EPR data collection and GO coordinated the project and wrote the final version of the article.

**Competing interests.** At least one of the (co-)authors is a member of the editorial board of Magnetic Resonance. The authors have no competing interests to declare.


**Acknowledgements.** We thank Dr. Adarshi Welegedara for the expression vector of calmodulin with amber stop codon and Dr. Angeliki Giannoulis and Ms. Yasmin Ben Ishay for some DEER measurements. Gottfried Otting thanks the Australian



Research Council, for a Laureate Fellowship (grant no. FL170100019). Ansis Maleckis thanks the European Regional Development Fund for funding. This research was made possible in part by the historic generosity of the Harold Perlman Family (Daniella Goldfarb). Daniella Goldfarb holds the Erich Klieger Professorial Chair in Chemical Physics.

**Financial support.** This research has been supported by the Australian Research Council (grant no. FL170100019 and DP210100088), the Australian Research Council Centre of Excellence for Innovations in Peptide and Protein Science (grant no. CE200100012) and the European Regional Development Fund (EDRF, PostDoc grant no. 1.1.1.2/VIAA/2/18/381).

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
