# Peer review of "Site-selective generation of lanthanoid binding sites on proteins using 4-fluoro-2,6-dicyanopyridine"

_Magnetic Resonance, 2022_

## Author Comment (AC1)

Below is our response to the comments made by Daniel Häussinger:

"1) $^1$H-DOSY spectra on the titration series of DCP-(L-Cys)$_2$ with YCl$_3$ should be performed in order to clarify the dimeric nature of the obtained shifts of the complex in slow exchange. (lines 378 ff.)"

Response: To address the stoichiometry of metal ion to DCP tag, we resynthesized the DCP-(L-Cys)$_2$ tag to measure its titration with yttrium more carefully by NMR spectroscopy. At 0.5:1 metal-to-tag ratio, the spectrum shows more NMR signals than expected for a simple equilibrium between metal-free and metal-bound tag, and an EXCSY spectrum shows that the signals of three different species exchange with each other at a rate of about 10 s$^{-1}$. A diffusion experiment indicates that one set of signals diffuses more slowly, in agreement with two DCP tags binding to a single metal ion. If any weak intensity remains for the free tag in the spectrum recorded at equimolar ratio of metal to tag, it is less than 3 % of the peaks of the integral of the 1:1 complex. At the ligand and metal concentrations of this sample (0.3 mM), we use
$K_d$ = [M][L]/[ML] = [0.03*0.3][0.03*0.3]/[0.97*0.3] = 0.00027 mM
to obtain a conservative estimate of the upper limit for the dissociation constant as 1 micromolar.

We also produced a new $^{15}$N-labelled sample of GB1 Q32C with DCP-(L-Cys)$_2$ tag. Titration of this protein sample with 0.6 equivalents of TmCl$_3$ yielded $^{15}$N-HSQC spectra that indicated a near-equimolar mixture of paramagnetic and diamagnetic protein signals. In a diffusion experiment performed with weak and strong gradients (the latter conducted to attenuate the signal intensities 2.3-fold) showed no distinguishable difference in attenuation of the Trp indol NH signals (which are resolved in the 1D $^1$H-NMR spectrum) between diamagnetic and paramagnetic species. Based on these data, there is no evidence for dimer formation in this sample. We do not understand why a greater concentration of lanthanoid ion led to sample precipitation for the GB1 construct.

In the revised version, we added spectra obtained for the DCP-(L-Cys)$_2$ tag and discuss the above in a new paragraph on pages 15–16, Fig. 6 and Fig. S16.

"2) The aforementioned titration experiments have to be carried out with different protein concentrations and should then allow the (at least approximate) determination of an affinity constant. The titration spectra should all be provided in the SI."

Response: 10-fold dilution of the protein did not significantly change the NMR spectrum. We cannot determine an affinity constant in this way. Instead, we

estimated an upper limit of the dissociation constant from spectra of DCP-(L-Cys)$_2$ as discussed in our response to point 1.

"3) In the list of crucial properties of an ideal lanthanoid binding site (lines 33ff) a high affinity between tag and metal is mandatory for a generally applicable system for structural work."

Response: We agree and added a fourth point in line 41.

"4) There is no information provided how the model of the complex for the rotamer libraries (line 166) was obtained (DFT?, force field?) and this holds also true for Figure S11. This should be included in the SI."

Response: For quite mobile tags such as the DCP tags of the present work, the location of the paramagnetic centre obtained by fitting the $\Delta\chi$ tensor rarely agrees with the modelled position. Arguably, the fitted location of the paramagnetic centre is more useful, as the utility of a $\Delta\chi$ tensor is determined by its capacity to predict PCSs. Accurate models of the metal ion position matter more in the case of DEER measurements. We used the metal position proposed by ChemDraw, which placed the metal ion 1.9 Å from the pyridine nitrogen and 2.3 Å from the thiazoline nitrogens. This is now stated in the legend of Fig. S13. As the tags performed less well in DEER experiments than previously published Gd-tags, we did not attempt to further refine the metal positions by DFT calculations or other energy minimisation algorithms.

"5) line 120 should read NMR: spectroscopy"

Response: thank you for noting, we corrected the typo.

"6) In the SI a proper characterization of compounds **2** to **5** (pages S3, S4) needs to be provided (at least NMR, MS)"

Response: NMR and MS data of compound **5** ($\beta$-cysteine) are now provided in the SI. Unfortunately, we do not have these data for compounds **2** to **4** and, as the organic chemist who synthesized these compounds no longer works in

chemistry, we cannot easily make them again. A synthesis protocol for β-cysteine was described already in 1956 by Birkofer and Birkofer without characterisation data. We could have referred to this publication, but we believe that it is useful to provide the more detailed description of our synthetic route. NMR data of compounds **2** and **3** were published by Bergeron et al. (1998) and Kim et al. (2014), respectively. We now cite all these publications in the Supporting Information.

Below is our response to the comments made by Nico Tjandra:

"The cysteine labeling reaction for a surface exposed cysteine is noted to complete overnight at RT, and the selenocysteine reaction is noted to have completed in 10 minutes (lines 175-180). This is stated to indicate that the reaction is selective for selenocysteine, but what is the actual timescale of the cysteine reaction? Is there a sufficient difference to avoid replacement of native cysteine residues?"

Response: Analysis by mass spectrometry indicated no reaction product for GB1 Q32C with FDCP after 10 minutes, incomplete tagging after 6 h and complete tagging after overnight reaction. This is now stated in lines 203 and 425.

"Additionally, buried cysteines are shown to label poorly or not at all with FDCP; is this the same for selenocys mutants?"

Response: We are reluctant to test this, as tagging buried cysteines would result in significant structural perturbations. Furthermore, the sensitivity of -SeH groups towards oxidation makes it difficult to handle samples containing multiple selenocysteine residues. In addition, the accessibility and, hence, reactivity of buried -SH or -SeH groups is expected to depend on the protein and its frequency of unfolding as measured by, e.g., exchange rates of backbone amides.

"The high salt requirement for the cyano group reaction with the free cysteine was noted. Was this to reduce aggregation of the protein in the reaction condition?"

Response: The high salt concentrations were used to limit unfavourable electrostatic repulsion between cysteine and the DCP. We don't know whether

this hypothesis is correct and therefore did not discuss it in the manuscript. Protein precipitation was not the issue.

"For the EPR experiments, several concerns were apparent.
First, in the NMR section, it is noted that metal: protein ratios > 0.6:1 resulted in significant protein precipitation, yet the EPR studies used stoichiometric or excess metal? Were there issues with precipitation in these cases, and could any of the peaks presented in the DEER distributions reflect protein aggregates?"

Response: We observed no protein aggregation when preparing the EPR samples. The EPR samples were of lower concentration than the NMR samples (0.1 vs 0.3 mM). In addition, the sample conditions were somewhat different due to the presence of 20 % glycerol in the EPR samples. The sensitivity towards precipitation may be protein specific.

"The authors indicate that multiple tags may be able to jointly coordinate additional lanthanides, but these results could also be explained by protein dimerization that leads to tags in close proximity. Would this problem be alleviated if the concentration of protein in the metal titration be lowered significantly?"

Response: Lowering the protein concentration, say, 10-fold, would entail recording NMR experiments 100 times longer to obtain the same signal-to-noise ratio, which is unpractical.

"As the authors acknowledge, the modulation depths of the EPR experiments used to check for tag mobility and lanthanide stoichiometry are very small. The authors are correct in their assertion that while these tags do not appear to be highly efficient for DEER experiments, they do appear very promising for paramagnetic NMR. Still, the pulsed EPR measurements are presented in this work as evidence for: the limited mobility of the lanthanide tags, the stoichiometry of the tags, and for the presence of expected distance distributions in each protein of interest. The very small modulation depths shown limit the ability to determine accurate widths for the distance distribution, and also the ability to draw conclusions regarding stoichiometry, as the vast majority of the ensemble is of unknown state. These experiments do indicate that at least some of the protein samples are tagged and in the

expected structural state, but this section would benefit from an additional method for validation of coordination stoichiometry and tag mobility."

Response: One aim of the DEER experiments was to obtain evidence for limited mobility of the tags (we do not claim that this aim was achieved!), another was to probe their general utility in DEER measurements. Notably, low modulation depths per se do not limit the ability to determine accurate distance distributions, provided the DEER data are sampled with sufficient signal-to-noise ratio and for a sufficiently long DEER evolution time. Indeed, the distances indicated by the main peak in each of our DEER distance distributions agreed with expectations from our models. As stated in the manuscript, problems easily arise from inaccurate titrations of the tagged protein with $Gd^{3+}$ ions and uncertainties in the stoichiometry with which DCP-1,2-aminothiol tags bind gadolinium. As shown in Fig. 4 and discussed in the manuscript, the DEER distance distributions obtained differed in appearance between samples titrated with $GdCl_3$ in 0.6:1 and 1.2:1 ratio of $Gd^{3+}$ to protein. Our new data on complex formation by the free DCP-(L-Cys)$_2$ ligand (Section 3.5) indicate that two ligands can indeed share a single lanthanoid ion, but whether this happens with a tagged protein would depend on the protein. The spectra shown in Section 3.5 indicate that free ligand persists in good concentration when the metal-to-ligand ratio is 0.5:1, indicating limited stability of the complex where two ligand molecules share a single lanthanoid ion. We added Section 3.5 for clarification.

The original manuscript addressed tag mobility by comparing the $\Delta\chi$ tensors obtained from PCSs and RDCs (Table 1), which showed clear evidence of tag mobility. Fortunately, PCSs generated by mobile tags can still provide solid structural information as discussed in the reference by Shishmarev and Otting, 2013.

Below is our response to the comments made by the editor:

"I attract your attention to the need to provide affinities, as well as the need to address the comments regarding DEER."

Response: Determining the metal binding affinities of the DCP tags is not straightforward. We tried isothermal calorimetry, which yielded no interpretable data. Terbium luminescence did not change in response to the presence of tag. As discussed in our new Section 3.5, NMR spectra of the DCP-(L-Cys)$_2$ tag indicate the dissociation constant to be less than 1 micromolar.

"In addition, please account for the following during your revisions to ensure reproducibility or clarity:

Several plasmids are mentioned but no identifiers are provided. Provide the identifiers for all your new plasmids so readers can request them easily."

Response: The genes of all proteins were in pETMCSI plasmids (described in Neylon et al., 2000). This is now stated in a sentence on line 80. We happily make all of them available on request. We tried generating specific names for each individual plasmid for the purpose of this article but found the text to become clumsy with no benefit.

"'prepared specifically for the present work by cloning the gene between the NdeI and EcoRI sites of the T7 vector pETMCSI'. This is unclear: please specify what gene in detail and provide the primers."

Response: The nucleotide sequences of the two genes referred to are now explicitly listed in Table S4 together with the mutation primers used.

"'0.2 mL/L trace metal mixture' either provide a reference or provide the composition of your stock solution if it departs from published protocols."

Response: We now re-state the reference (which is Klopp et al.).

"Specify that the Ni-NTA column is His GraviTrap TALON when first mentioning His-tag purification since you later refer to this column but had not provided its description."

Response: Thank you for pointing out the inconsistency. We now specify the columns used more clearly in lines 92 and 109.

"For FDCP tagging, please specify how the buffer was exchanged in detail (specifying the column if using a column, specifying the number of cycles if using dilution/concentration, etc.)"

Response: The details are now provided in lines 124-125.

"Please verify that all acronyms and abbreviations are defined. For example, RQ-SLIC is not defined."

Response: this is now defined in line 72. The abbreviation DTT is now defined in line 108.

"Please specify the vendor when referring to a product, including identifiers if several options are available. This condition is in general respected but not always. For example, QuickChange needs the vendor information (assuming you indeed used a kit)."

Response: In the case of RQ-SLIC and QuikChange, we used our own in-house protocol and reagents as described in the reference by Qi and Otting, 2019.

"There are a few typos or grammatical mishaps that occasionally distract readers in an otherwise well-written manuscript. Please give this manuscript another round of proof reading after modifying it.

Examples "useful for the structural characterisation of protein",  "Samples were produced of nine different proteins".

"Standard expression" --> "Expression in rich media" or "Expression for unlabeled proteins"

Response: we amended the wordings.

---

## Referee Report (RR1)

In the current form, the manuscript by Tharayil et al. is suitable for publication. The upper boundary for the metal affinity of the tag of 1 μM is very helpful, as well as the performed DOSY and EXSY experiments, that indicate that a dimerization of tagged proteins does not contribute significantly. The very useful manuscript should find many readers in the broad community of MR.

---

## Author Response (AR2)

Compared to the accepted version of the manuscript, the version uploaded today contains some very minor corrections:

- Added a second doi for NMR data uploaded to Zenodo (namely for spectra added in the revised version) – line 453,
- Corrected instances of SARS-2 to SARS-CoV-2 (which is the correct name and used also in the titles of the references cited, as pointed out by one of the co-authors),
- Fixed a typo by putting brackets around the first occurrence of PpiB – line 75,
- Removed blank spaces and hyphens in compound names – line 22 and 57.